# A Multistage Analysis of Asphalt Binder Nanocrack Generation and Self-Healing Behavior Based on Molecular Dynamics

**DOI:** 10.3390/polym14173581

**Published:** 2022-08-30

**Authors:** Haoping Xu, Wenyuan Xu, Xuewen Zheng, Kai Cao

**Affiliations:** 1School of Civil Engineering, Northeast Forestry University, Harbin 150040, China; 2College of Civil Engineering and Architecture, Zhejiang University, Hangzhou 310058, China

**Keywords:** asphalt binder, nanocrack, molecular dynamics, self-healing

## Abstract

In order to study the characteristics and laws of nanocrack generation and self-healing behavior of asphalt materials under tensile action, the molecular dynamics (MD) method was used to simulate the continuous “tensile failure—self-healing” process, and this study remedies the shortcomings of existing experimental and observational methods. It is found that the MD-reproduced formation process of asphalt binder nanocrack contains four stages: “tensile extension”, “nanocrack generation”, “crack adding, expanding and penetrating” and “cracking failure”. The influence of tensile conditions on the tensile cracking simulation of an asphalt binder model was analyzed, and it was found that low temperature and high loading rate would increase the tensile strength of the asphalt binder model. In addition, the MD-reproduced healing process of asphalt binder nanocracks can be divided into four stages: “surface approach”, “surface rearrangement”, “surface wetting” and “diffusion”, which is similar to the healing process of polymers. Finally, from the perspective of energy change, the change rule of dominant van der Waals energy in the self-healing process was studied. Based on the existing research, the influence of damage degree on the healing performance of asphalt binder and its mechanism were further analyzed. The research results further enrich the theoretical research on microlevel cracking and healing of asphalt materials, and have certain theoretical value for the further development of self-healing asphalt materials.

## 1. Introduction

Asphalt binder is a main building material with complex components but good viscoelasticity. The pavement built by it often has a flat surface, is jointless and has comfortable driving, wear resistance, low noise and other strong points. Therefore, it has been widely used in transportation [1]. According to the survey, more than 90% of the pavements in modern cities are composed of asphalt materials [2], and how to avoid or repair the cracking of asphalt pavement has always been a hot issue that many researchers are concerned about.

In the early stage of cracks in the asphalt pavement, the nanocracks inside the asphalt material have little impact on the smoothness and driving comfort of the pavement. However, in a humid environment, these nanocracks will expand and cause water damage and other diseases, which will seriously affect the performance and longevity of the pavements [3,4]. Current researchers tend to use two methods to avoid asphalt pavement cracking failure, by developing stronger asphalt pavement materials or studying the healing properties of asphalt pavement materials.

The self-healing mechanism of asphalt binder was originally developed from the polymer self-healing theory. In 1980, Wool and O’Connor [5,6] first proposed that the self-healing of polymer materials includes the following five stages by studying the healing phenomenon of polymer materials: (1) surface rearrangement, (2) surface approach, (3) wetting, (4) diffusion and (5) randomization. Figure 1 shows a diagram about the self-healing stages of polymer materials proposed by Liang Bo [7]. In this figure, it is assumed that the blue dot and gray dot represent the molecules at both ends of polymer materials after complete cracking. Observing this figure, we can find the characteristics of the five stages in the healing process of polymer materials: the surface rearrangement and the surface approach stage occur almost at the same time, and after this, there is still a certain void area at the two interfaces of the crack model; the wetting process shows that the void area in the interface at both ends of the crack gradually decreases and disappears; the initial diffusion of molecules between the interfaces is followed—at this time, the phenomenon of molecular diffusion is concentrated in the vicinity of the crack interface, and there is no molecule from the other end in the part far away from the crack interface; finally, there is a comprehensive diffusion and rearrangement. At this stage, it can be clearly observed that molecules of different colors from both ends have been fully diffused into the whole model. Theoretically, when this stage is reached, the original crack interface will completely disappear. At this time, the strength of the material can reach the strength of the raw material. Inspired by research on polymers, research on the self-healing mechanism of asphalt materials is also developing rapidly.

Researchers believed that the self-healing ability of asphalt materials can be regarded as a complex process of self-healing of stiffness and strength, which can occur during the damage process, at rest or during high temperature [8]. Subsequently, a large number of studies and tests confirmed that asphalt materials have a certain self-healing ability to resist fatigue damage [7,9,10]. Daquan Sun joined two asphalt fragments together with a small force and observed their healing phenomenon with a fluorescence microscope. The self-healing process of the asphalt fragments could be divided into a wetting stage and a molecular diffusion stage. It was found that the crack self-healing speed was slow in the wetting stage, but significantly increased in the molecular diffusion stage [11]. Shihui Shen used field-emission scanning electron microscope to verify that the asphalt binder has self-healing ability, and then simulated the asphalt binder self-healing process by molecular dynamics. By analyzing the simulation data, the asphalt self-healing is divided into two stages: short-term healing and long-term healing. Short-term healing is mainly responsible for the recovery of modulus, while long-term healing can achieve the complete recovery of fatigue performance [12]. Quan Lv divided the asphalt binder self-healing process into three stages: gathering, moving and rounding by analyzing the CT scanning images of the asphalt materials’ self-healing process [13]. Kim [14], Little [15], Hammoumm [16] and other scholars have confirmed the self-healing ability of asphalt materials in the laboratory or in the actual pavement.

At present, the self-healing mechanism of asphalt materials mainly includes crack surface free energy theory [17], crack surface molecular diffusion theory [18] and capillary flow theory [19,20]. The factors affecting the self-healing ability of asphalt materials mainly include the chemical composition of asphalt materials, environmental impact, intermittent time, asphalt modifier, etc. Due to the complex composition of asphalt materials and the limitations of instruments and technologies, it is very difficult to analyze the microscale of the generation, evolution and healing process of nanocracks in asphalt materials in the early stage [7,9]. Molecular dynamics (MD) provides a new method for studying the micromechanism of asphalt materials. MD simulation is a computer simulation of a large number of molecular motions based on the physical principles of atoms and molecules. It is a calculation method based on statistical mechanics and thermodynamics to simulate the interaction and behavior of various atoms and molecules under certain conditions. MD simulation was first applied to the study of protein molecules in biology, and was later developed in petrochemicals. It was used to study the properties of oil and gradually applied to the asphalt field [21]. Tengjiang Yu used molecular simulation technology to divide the self-healing process of asphalt materials into three stages: turbulent, distance self-healing and strength self-healing by analyzing the density change curve in the self-healing process, and proposed that the second stage is the most influential process of asphalt self-healing [22]. Daquan Sun created an artificial crack in the asphalt MD model to simulate the self-healing ability of the asphalt material model, and found that the self-healing ability of the asphalt materials increases with the temperature; the self-healing ability of SBS-modified asphalt is better than that of matrix asphalt. In addition, the relationship between activation energy, preexponential factor and molecular self diffusion in MD simulation is qualitatively consistent with the observed results of the DSR experiment [23]. Liang He simulated the molecular diffusion behavior of aged asphalt, SBS-modified asphalt and virgin asphalt in the self-healing process through molecular simulation technology. The phenomenon shows that the diffusion coefficient of the asphaltene molecule is the lowest and that of the saturated component is the highest in the self-healing process of asphalt materials. The self-healing behavior of asphalt materials is mainly based on the van der Waals force between molecules. The aging of asphalt binder molecules reduces the diffusivity of the asphalt binder model, while SBS additives indirectly improve the diffusivity of asphalt binder [24]. All the findings based on MD simulation are helpful to further understand the failure process of materials at the microscale, which cannot be observed under the conventional experimental conditions of asphalt materials.

In this paper, a molecular model of asphalt binder is first developed and then a constant-rate uniaxial stretching is performed to analyze the mechanical properties of the model after verifying its accuracy in various aspects. During the uniaxial stretching session, multiple gradients of temperature and stretching rate will be used as variables to analyze their effects on the tensile failure of the asphalt model. The resulting tensile failure model will then be used as the initial model for the next stage of studying the self-healing behavior of the asphalt materials rather than by inserting artificial cracks into the two asphalt models. Finally, the influence mechanism will be investigated based on the existing phenomenon of the influence of the asphalt damage degree on self-healing ability, in order to further understand the self-healing ability and mechanism of asphalt binder.

## 2. Research Methods

### 2.1. Asphalt Binder Molecular Model Selection

Asphalt binder is currently one of the main road materials; it is a mixture of com-ponents containing a large number of hydrocarbons of different molecular weights, a small number of heteroatoms (such as nitrogen, oxygen and sulfur) and metal atoms. The interaction between atoms determines the physical and chemical properties of asphalt binder.

There are two kinds of models adopted by researchers for asphalt binder molecular dynamic simulation, which are average molecular model and multicomponent Model. However, it is difficult to describe various properties of asphalt binder with one molecule when using average molecular model. Therefore, the 4-component, 12-molecule model in multicomponent model is used as the object of asphalt binder molecular simulation. The 4-component theory of asphalt binders proposed by L.W. Colbert classifies complex asphalt molecules into four groups: saturate, aromatic, resin and asphaltene [25]. Compared with the average molecular model, the multicomponent model can obtain more accurate results in the simulation of density, expansion coefficients, and equivalent compression rates. It can also study the impact of different components in asphalt binder molecular dynamics.

With the continuous development of molecular simulation theory, many researchers have conducted a lot of work around the establishment of a more accurate asphalt model. According to the ratio of four components, the mass percentage of carbon, hydrogen, oxygen, nitrogen and sulfur elements, the atomic H/C ratio and the percentage of aromatic hydrocarbons and alkanes, Li proposed three kinds of asphalt four-component models in SHRP schemes. Compared with the previous asphalt models, these three asphalt molecular models are more reasonable and closer to real asphalt binder in terms of physical properties and thermodynamics [26].

The asphalt binder model used in this study is shown in Figure 2. These include asphaltene components: asphaltene-phenol, asphaltene-thiophene, asphaltene-pyrrole; saturated components: squalane and hopane; aromatic components: PHPN (perhydrophenanthrene-naphthalene) and DOCHN (dioctyl-cyclohexane-naphthalene); and resin components: quinolinohopane, thioisorenieratane, benzobisbenzothiophene, pyridinohopane and trimethylbenzeneoxane. The types and numbers of molecular models for AAA-1 asphalt binder are shown in Table 1. This molecular model has been repeatedly validated in previous studies and has been used to analyze the diffusion [27], self-healing [28,29], rejuvenation [29] and adhesion [28] of asphalt materials at a nanoscale. Therefore, this model is suitable for molecular dynamics research on the generation and self-healing of nanocracks in asphalt binder systems in this paper.

### 2.2. Details of the Molecular Dynamics Simulation

In this study, large-scale atomic/molecular massively parallel simulator (LAMMPS) [30] was used to simulate the direct tensile failure and self-healing behavior of asphalt materials at the nanoscale, and the PCFF (polymer consistent forcefield) was used to describe the forces between individual atoms. The PCFF is a second-generation forcefield based on the improvement of the CFF91 force field, which contains very complex potential energy forms and parameters. The PCFF force field has been extended in the scope of application, mainly for the simulation of polymers and organic materials. Now the parameters of this forcefield have been modified and supplemented many times, and it has been able to simulate other materials such as inorganic materials and metals. The PCFF forcefield has been used in many asphalt MD studies and accurately predicts the relevant material properties, so it is suitable for the molecular dynamics study of asphalt binder systems in this study [2,31]. Visual Molecular Dynamics (VMD) [32] and Open Visualization Tool (OVITO) [33] were used for visual processing of the model.

After establishing 12 representative molecules of AAA-1 asphalt binder, the combined model of AAA-1 asphalt binder was established by Packmol; the initial model size was 140 Å × 140 Å × 140 Å, and the density was approximately equal to 0.1 g/cm^3^. The simulation steps used in this paper are shown in Table 2. Figure 3 shows the picture of the asphalt binder model after these processes.

### 2.3. Tensile Simulation of Asphalt Binder Model

After the stable asphalt binder model is obtained, this asphalt binder model is then subjected to tensile simulation. The loading method adopted in this study is to assign a constant deformation to both ends of the asphalt binder model atoms, which can be regarded as a tensile deformation of the entire asphalt binder model in one direction with a constant velocity. When the model starts to be stretched, the atoms at the two ends move to the outside first, while the atoms at the inside move under the influence of the molecular force generated by the movement of the atoms at the two ends. The whole process will be adjusted to the specified temperature using the NVT ensemble and apply a constant outward deformation in the z-axis direction.

In order to study the effects of tensile rate and tensile temperature on the tensile simulation results of asphalt binder model, five temperatures (−25 °C (248.15 K), 0 °C (273.15 K), 25 °C (298.15 K), 50 °C (323.15 K) and 75 °C (348.15 K)) were selected to simulate and analyze the tensile properties of asphalt binder models from low temperature to extreme high temperature [34]. In addition, Hao Wang used MD to simulate and analyze the bond strength of the asphalt–aggregate interface. He studied the effect of loading rate in the separation process of the asphalt–aggregate interface by applying a constant tensile load from 50 m/s to 0.1 m/s to the asphalt–aggregate model [35]. Due to the limitation of hardware and MD time scale, the strain rates adopted in tensile MD simulation are often higher than the actual strain rate. In this study, four strain rate conditions of 50 m/s, 25 m/s, 10 m/s and 5 m/s are selected to study the influence of strain rate on tensile simulation of asphalt binder model. The tensile simulation will stop when the model strain reaches 100% on the z-axis; Figure 4 shows images of asphalt binder model under different strains at 25 °C and 10 m/s.

### 2.4. Self-Healing Simulation of Asphalt Binder Model

The purpose of this part is to analyze the self-healing behavior of asphalt binder model and the data changes during this process. The asphalt binder nanocrack model obtained by tensile simulation in the previous step will be used as the initial model for multistage analysis of asphalt binder healing model in this part. During the self-healing simulation of asphalt binder model, NPT ensemble was used throughout the simulation process; Nose-Hoover thermostats and Berendsen were used to control the temperature and pressure, respectively; and periodic was selected as the boundary condition. The time step was 1 fs, the pressure was 1 atm and the cut-off distance was 15.5 Å.

When previous researchers used MD to study the self-healing behavior of asphalt, most of them chose to artificially insert a vacuum zone of a specific width between two periodic asphalt models (as shown in Figure 5a) to simulate the nanocrack in asphalt binder [23,24,36]. Due to the use of asphalt model with periodic structure, the asphalt model surface on both sides of the nanocrack has a more suitable and stable structure. This not only greatly weakens the phenomenon of surface rearrangement in the healing process, but also forms a consistent contact surface after the two asphalt models make contact with each other in the simulation process, thus speeding up the self-healing progress of the asphalt model. In this study, the previous asphalt binder tensile failure model was used as the initial model for the asphalt binder self-healing simulation (as shown in Figure 5b). Observation of Figure 5b shows that the initial model of asphalt binder self-healing simulation selected in this study does not have a high degree of compatibility between the two crack surfaces. To some extent, using this model as the initial model to simulate the self-healing process can more accurately represent the initial model of asphalt binder nanocrack self-healing.

## 3. Results and Discussion

### 3.1. Model Verification

In order to verify the reliability of the asphalt binder model, Table 3 presents the simulated and experimental data of the (i) density, (ii) glassy transition temperature, (iii) solubility parameters and (iv) diffusion coefficient.

As an important index in molecular simulation, the density of the asphalt binder model at 298.15 K is 0.981 (g/cm^3^), which is lower than the density of 1.01–1.04 (g/cm^3^) obtained in the experiment, which may be due to the excessive complexity of the asphalt binder components, and the asphalt binder model constructed with 12 representative molecules cannot fully represent the complex composition of the actual asphalt materials.

Glass transition temperature (T_g_) is the temperature corresponding to the material’s transition from a glassy state to high elastic state. Many properties of the material will change sharply near the glassy transition temperature. The specific volume can be obtained by using a series of NPT ensemble at different temperatures on asphalt binder models, and the glass transition temperature could be estimated by observing specific volume–temperature curves. The glass transition temperature of this model is 267.7 K, which is close to the experimental data.

The solubility parameter is a physical parameter to measure the compatibility between substances. If the solubility parameters between two substances are closer, the two substances will be easier to mix. The calculation formula of solubility parameters is shown in Equation (1):(1)$$\delta =\sqrt{\frac{E_{coh}}{V}}$$
where *E_coh_* is the cohesive energy and *V* is the model volume. The physical significance of the solubility parameter *δ* can be regarded as the square root of the cohesive energy density of the material.

### 3.2. Tensile Simulation and Analysis

In this part, the tensile simulation phenomenon of asphalt binder with 25 °C, 10 m/s, will be used as the analysis object. The stress–strain curve, interaction energy change and cracking volume change of the system are mainly analyzed. The purpose is to simulate the tensile fracture process of asphalt materials at the nanosize and analyze the generation and evolution of asphalt binder nanocracks. Figure 6 shows the stress–strain curve of the asphalt binder model and the change of interaction energy when the tensile strain is from 0% to 100%.

It can be seen from Figure 4 that the initial model is gradually separated into two model fragments under the influence of constant tensile rate. In addition, it can be found in Figure 6 that the stress in the early stage of tensile simulation increases rapidly with the increase in strain until the stress–strain curve reaches the peak value (117 MPa) when the strain reaches 11%, and then begins to decline and approaches 0 MPa when the strain is about 60%; when the strain is less than 17%, the interaction energy curve shows an upward trend, and the change rate of the interaction energy increases rapidly before the peak stress (11% strain), but slows down after 11% strain. It is worth noting that when the model strain reaches 17%, the interaction energy begins to decline slowly, which is most likely caused by the rearrangement of atoms at the crack interface. At the same time, it can also explain why the increase rate of the interaction energy slows down at the 11–17% stage.

Figure 7 and Figure 8 show the distribution function of density on the z-axis of the model and the change data of each volume under different strain degrees. The density distribution curves shown in the figure are divided into multiple parts to facilitate the observation of the difference (except for 0% strain condition, 5%, 10%, 15%, 20%, 30%, 45% and 60% strains are selected as the time points, and each density distribution map is obtained by averaging all the density curves of the previous 5% strain). The crack volume was calculated by OVITO’s Construct Surface Mesh module, which generates a geometric description of the outer and inner boundaries of an atomistic solid in terms of a triangulated surface mesh. Aside from visualization purposes, the geometric description of the surface is also useful for quantitative measurements of the surface area and the solid volume and porosity of an atomistic structure. It should be noted that the radius of the probe sphere selected in this paper is larger than the commonly used radius when calculating the crack volume of the model (the minimum probe radius is selected when the pore volume inside the initial model is 0, which is 3.7 Å in this paper) to exclude the influence of the original small pores in the initial model. Selecting a probe sphere with a larger radius will lead to a larger volume of asphalt completely equal to the total volume at the beginning of the simulation, and the new pore area in the model will be regarded as a nanocrack.

By observing Figure 7 and Figure 8, it can be seen that when the model strain is 0–5%, the increment of solid volume occupied by asphalt is equal to the increment of total volume, and the system is in the volume expansion stage caused by stretching. The fluctuation of density curve indicates that there are a small amount of pore areas in different positions inside the model, but these pore areas are not considered as cracks because they do not meet the requirement of the probe sphere. These phenomena indicate that the asphalt binder model is in the stage of “tensile extension” during this period. Subsequently, when the strain is between 5–10%, the cracking volume begins to rise from 0, which means that there are pore areas in the asphalt binder model that can be detected by the probe sphere. The asphalt volume continues to increase in this stage, but its rate starts to decrease, while the crack volume increases at a slower rate compared to the subsequent stages. This means that although detectable cracks appear inside the asphalt model at this stage, the asphalt binder model still has a certain elongation capacity and the volume of solids that grows by elongation is greater than the volume that decreases by cracks, so this stage can be considered as the asphalt binder model being in the stage of “nanocrack generation”. When the strain is at 10–20%, the asphalt density distribution curve begins to fluctuate significantly as the strain rises, and a large and a small density trough appears at 40% and 60% of the z-axis, respectively, which indicates the appearance of new fine cracks around the main cracks at 40% of the z-axis. The asphalt volume value continues to decrease while the crack volume increases rapidly, and the rate of crack volume increase is further accelerated at this stage. This indicates that the tensile extension capacity of the asphalt binder model has reached its limit at this point, and new detectable cracks continue to appear, while the existing cracks within the model continue to expand as they stretch. These phenomena indicate that the asphalt binder model has entered the stage of “crack adding, expanding and penetrating”. When the strain is in the 20–100% stage, the analysis of the asphalt density distribution curve shows that the density value further decreases at the location of the previous density trough, and all the asphalt binder molecules keep gathering towards the ends of the z-axis. At this time, the asphalt volume decreases more and more slowly, and the crack volume growth rate is close to the total volume growth rate, while the density distribution curve indicates that the cracks at 40% and 60% of the Z-axis position continue to expand after merging with each other, so the reason for the slow decline of the asphalt volume curve after excluding the effect of the appearance of new cracks may be caused by the rearrangement of atoms on the crack surface. This deduction can also explain the slowing down of the interaction energy increase in the strain range of 12–17% shown in Figure 6, and the slow decrease in the interaction energy after the strain is greater than 17%. “The cracking failure” will be considered as the last stage in the tensile simulation of the asphalt binder model in this study

In this part of the simulation, the increase in tensile strain at the beginning of the asphalt binder model tensile simulation leads to the appearance of small pores inside the model, which can be regarded as the generation of nanocracks, but the asphalt binder model in this stage is in a relatively healthy state; the asphalt volume value, interaction energy and tensile strength are in an increasing trend. However, with the increase in tensile strain, the nanocracks inside the model keep expanding and generating, some of the nanocracks merge with each other, and the cohesion of the asphalt binder model decreases until the entire asphalt binder model is divided into two independent model pieces, which means that the asphalt binder model undergoes four stages of “tensile extension”, “nanocrack generation”, “crack adding, expanding and penetrating” and “cracking failure” before complete destruction and crack damage formation.

### 3.3. Effect of Temperature and Tensile Rate on Tensile Simulation

In previous studies, Hao Wang [35] analyzed uniaxial tensile simulations of asphalt–aggregate models using molecular dynamics methods and concluded that model size, loading rate and temperature have some influence on tensile strength results, which is similar to the findings of Ye-shou Xu [38], D. Hossain [39] and others for tensile simulations of rubber models and polymer models. However, the effect of different loading conditions on the simulation results during uniaxial tensile simulations regarding asphalt binder models has not been clarified.

Therefore, this section further investigates the effect of loading rate and temperature on the evolution of microcracks in asphalt binder by analyzing the differences of tensile failure process of asphalt binder models under different temperature and deformation rate conditions during the tensile simulation. In this paper, the selected temperature conditions are −25 °C, 0 °C, 25 °C, 50 °C and 75 °C, and the selected tensile rates are 50 m/s, 25 m/s, 10 m/s and 5 m/s. The tensile rate is 10 m/s when analyzing the effect of the temperature factor, and the temperature is 25 °C when analyzing the effect of the tensile rate factor. All tensile processes were stopped when the z-axis strain of the model reached 100%. Figure 9 and Figure 10 show the variations of stress and interaction energy for different temperature and tensile rates conditions.

Observing Figure 9, we can find that the trends were similar for the different stress–strain curves in the selected temperature range; as the temperature changed from −25 °C to 75 °C, the stress–strain curve peak coordinates were (8.852, 143.58), (9.278, 138.77), (10.823, 115.53), (11.339, 102.05), (11.873, 85.72). The tensile stress extremes appear later and later, while the tensile stress extremes become smaller and smaller; in addition, observing the interaction energy change curve, it is found that the extreme value increases simultaneously with the increase in temperature. This may be due to the increase in free volume and the growth of interatomic distances within the model at high temperatures, which in turn leads to a decrease in the interaction energy and cohesion capacity of the model. These two curves indicate that the uniaxial tensile strength of the asphalt binder model decreases with increasing temperature, but its tensile elongation capacity also increases relatively. (Note that this part of the interaction energy is always negative, which means that the larger the value of the interaction energy change, the closer the interaction energy between model atoms is to zero, and the smaller the model cohesiveness).

Subsequent observation of Figure 10 shows that the extreme values of the stress–strain curves increase as the loading rate increases, but they occur at very similar points in time, and the extreme values of the interaction energy also show an increasing trend and occur at a later time. In the previous section for the analysis of energy change and crack volume during the tensile simulation of the asphalt binder model, it was indicated that the model would experience rearrangement of crack surface atoms to form a lower energy structure later in the simulation, and the rearrangement of surface atoms would lead to a slow decrease in the values of interaction energy. In this part of the study, the energy change curve at high loading rate shows a higher peak and a later peak appearance. Therefore, after combining the energy change curves in Figure 10 with some conclusions from the previous part of the study, it can be concluded that the high loading rate slows down the rearrangement of the crack surface atoms, thus leading to a later appearance of the interaction energy peak, which also indicates that the energy analysis of the model is more accurate at low loading rates.

### 3.4. Multistage Analysis of Self-Healing Behavior of Asphalt Binder Model

The theory of self-healing of asphalt binder was first developed from the theory of polymer self-healing, so this section will analyze asphalt binder self-healing with reference to the theory of polymer self-healing research and analyze the different situations and characteristics of the asphalt binder self-healing process in stages with reference to polymer model stages.

Figure 11, Figure 12 and Figure 13 show the variations of multiple properties and model structure during the self-healing simulation. These figures demonstrate a typical asphalt binder model self-healing process.

By observing and comparing the changes in the polymer self-healing process shown in Figure 1 with the asphalt binder healing process model shown in Figure 12, it can be found that the asphalt binder model has similar stages in the self-healing process as the polymer model: the surface approach stage, which is mainly characterized by the approach of model fragments to each other and the short duration of this stage; the surface rearrangement stage, which is considered as the process of the slow movement of the crack surface atoms from the unreasonable structure due to tensile damage until the formation of a lower energy structure, and is therefore a stage highly relevant to the existence of cracks in asphalt binder model; the surface wetting stage, which generally exists when the asphalt binder model fragments are close to each other and the surfaces can not be completely fit. At this time, the pore areas on the surface of the model fragments are not coincident, but will gradually disappear as the surface wetting. However, this stage is often ignored or merged into the surface approach stage due to its small time span or unreasonable asphalt binder model construction; and the diffusion stage, which as one of the stages in the self-healing process of asphalt binder has the greatest impact on the recovery of material strength [40]. The diffusion stage is essentially where the surface molecules of one asphalt binder fragment continue to move after experiencing mutual entanglement in the surface wetting stage, even crossing the original crack interface into the other asphalt binder fragment. Theoretically, after a long enough period of diffusion of the two model fragments, the original crack interface of the material will completely disappear and the strength will return to the pre-cracking level.

As can be seen from Figure 11, the starting point of the volume curve and density curve is located at a very large and very small value, respectively, due to the existence of cracks in the self-healing process of the asphalt binder model. The density distribution curve shows that a large number of molecules are gathered at both ends of the z-axis (tensile axis) and there is a vacuum zone in the middle part. As the healing simulation proceeds, the density curve and volume curve change rapidly in the first 20 ps. It is worth noting that the middle of the density distribution curve still has a region with a value of 0. Combined with the model diagram, we can conclude that the two ends of the model are approaching each other but not in contact, and the asphalt binder self-healing model is at the stage of surface approach and surface rearrangement. Subsequently, when the time reaches 30 ps, we can see that the middle part of the density curve starts to rise from 0. However, Figure 13 shows that the two ends of the asphalt binder model are still separated and not in contact, so the model is still in the surface rearrangement and surface approach stage. When the simulation time reaches 35 ps, Figure 13 shows that a blank area appears in the crack region of the asphalt binder model, which means that the asphalt binder model is connected for the first time and the wetting stage starts slightly before this time point, while the diffusion stage will start slightly after this time point, so the model is in the surface rearrangement, surface approach and surface wetting stages at this time. When the simulation time reaches 100 ps, by observing Figure 11a, it can be found that the volume and density values of the asphalt binder model begin to stabilize at this time, which means that the stage of interface approach has ended. By observing the asphalt binder model, we can find that there is still a certain volume of void area in the interface, so the asphalt binder model is in the stage of surface rearrangement, surface wetting and diffusion at this time. During the period from 100 ps to 600 ps, the density of the model increases very slowly, the interatomic interaction ability increases due to the proximity, and the volume of the voids in the asphalt model binder gradually decreases, at which time the crack volume in the model is calculated to be less than 1% using a probe of size 3.7 Å. This indicates that the two originally separated asphalt binder fragments are in a close-contact state and implies the end of the surface wetting stage, when the asphalt binder model is only in the diffusion stage.

After analyzing the self-healing state of the asphalt binder model, Figure 14 shows the distribution of the four stages of asphalt binder self-healing during the “tensile-self-healing” simulation. It is noteworthy that the asphalt binder self-healing stages in this study are overlapping structures rather than end-to-end, and this overlapping structure will be more suitable for the analysis of asphalt binder self-healing when the crack surfaces do not fit together.

### 3.5. Relationship between Self-Healing Behavior and Energy of Asphalt Binder Model

In MD simulations about bituminous materials, researchers generally divide the energy of the whole system into two major parts: the bond energy present within individual molecules and the non-bond energy present between molecules. Taking the asphalt model binder healing process at 298.15 K, 1 standard atmospheric pressure as an example, the E_bond_ = bond energy, E_angle_ = angle energy, E_dihed_ = dihedral energy, E_imp_ = improper energy, E_vdwl_ = van der Waals pairwise energy (includes E_tail_), E_coul_ = Coulombic pairwise energy, E_long_ = long-range kspace energy inside the system throughout the process will be determined as the main observation target, recorded in Figure 15. It is easy to find that E_vdwl_ plays a dominant role in the self-healing process of the asphalt binder model.

In addition, controlling the tensile simulation time at the same temperature and tensile rate can result in multiple asphalt binder models with different strain levels, which can be regarded as differences in crack width or crack volume. These models with different degrees of damage can be used as the initial models for studying the effect of crack size on asphalt binder self-healing properties. Previous studies have found that crack width or crack volume play a very important influence on the self-healing ability of asphalt binder [41,42], and this study will further investigate the mechanism based on this phenomenon in terms of energy change. Figure 16 shows the variation curves of the dominant energy E_vdwl_ in the self-healing process of asphalt binder models with different damage degree conditions. The trend of the E_vdwl_ change of the model in Figure 16 is approximately the same for all conditions, which indicates that the degree of damage does not affect the distribution of stages in the self-healing process. It is further inferred that the conclusions of the multistage analysis of the self-healing behavior of the asphalt binder model in Chapter 3.4 are applicable to the asphalt binder self-healing process under different damage conditions. In addition, a similar inflection point can be found for all the curves in the images. This inflection point splits the entire curve into a curve A, which falls rapidly over a short period of time, and a curve B, which falls slowly over a long period of time. Due to hardware limitations, the self-healing simulation time in this study is short, and only a small period of time before and after the appearance of the inflection point is recorded in Figure 16, which means that the actual time span of curve B is much larger than that shown in the figure.

Previous researchers believed that after a long enough interval, small cracks within the material can heal themselves and the material properties can be restored to their pre-damage properties, so self-healing is considered the inverse process of cracking [22,43]. The initial model of healing used in this section is obtained by simulating the same model for different times of stretching, so it is not difficult to deduce that for a long enough simulation time, the van der Waals energy values of all healing models will be close to the van der Waals energy values of the asphalt binder model before the stretching failure E. Due to the difference in damage degree of asphalt binder models at the beginning of the healing simulation, the initial van der Waals energy E_1_ of each healing model is also different. In addition, if the van der Waals change in curve A can be regarded as ΔE_2_ and the van der Waals change in curve B as ΔE_3_, these values should satisfy Equation (2).
E = E_1_ + ΔE_2_ + ΔE_3_,(2)

Formula (3) is obtained after transposing Formula (2):ΔE_3_ = E − E_1_ − ΔE_2_,(3)
where the van der Waals energy E of the asphalt binder model before tensile failure is a fixed value (−7411.25699 kcal/mol in this study). The values of E_1_ and E_2_ for different degrees of injury can be obtained by observing Figure 6 and Figure 16; thus, Table 4 is obtained by calculation.

It can be observed in Table 4 that the larger the degree of damage, the smaller the value of E1 for the asphalt binder model. This phenomenon may be caused by the different degree of interfacial molecular rearrangement in the initial healing model under different degrees of injury (discussed in Section 3.2). Subsequently, it can be found that the values of ΔE_2_ increase with the degree of damage, but the values of ΔE_3_ were very close. These phenomena indicate that the van der Waals energy values of asphalt binder models with different degrees of damage are different at the beginning of self-healing, but the values of van der Waals energy are close for all models after self-healing through the stage A of the curve. Since van der Waals force is a kind of interaction force between molecules, and the number of atoms and the cut-off distance did not change during the simulation, the structure and density of all models should be similar at the end of the self-healing of the curve A stage. Because the temperature and pressure conditions for the self-healing of asphalt binder models with different damage levels are the same, the ΔE_3_ values are close to each other, and the structure and density of the models are similar after the end of the curve A stage, it can be speculated that the damage level cannot significantly affect the time required for the healing of the curve B stage.

Figure 17 records the appearance time of the inflection point and the crack volume ratio under different damage conditions, where the appearance time of the inflection point can be regarded as the self-healing time consumed by the curve A stage. It can be seen that with the increase in damage degree, the increment of time required by curve A under the adjacent damage degree gradually increases. It can be expected that when the damage degree rises to a certain value, the healing time will rise rapidly to a huge value, and then the asphalt binder model will be difficult to self-heal. It is also observed that the crack volume ratio values for the asphalt binder models with different degrees of damage are less than 1% at the end of the curve A stage, which means that the asphalt binder models are very tightly fitted together. After combining the conclusions of Section 3.4, it can be further inferred that the inflection point of the intersection of curve A and curve B occurs around the end of the surface approach, surface rearrangement and surface wetting stages. At the same time, it means that the damage degree can affect the self-healing performance of the asphalt binder model by changing the time and efficiency of the surface approach, surface rearrangement and surface wetting.

In this study, based on the verification of the effect of asphalt crack volume on asphalt binder self-healing behavior, the influence mechanism was further analyzed from the perspective of energy change. It is shown that van der Waals energy plays a dominant role in the self-healing process of asphalt binder models, and different damage levels do not affect the distribution of stages in the self-healing process. Lower damage levels improve the efficiency of surface approach, surface rearrangement and surface wetting and shorten the elapsed time of these stages, thus showing a stronger healing ability.

## 4. Conlusions

(1)The tensile failure of asphalt binder was simulated using MD, and the process of crack generation, expansion and complete formation was analyzed, which could not be observed in experiments. It was found that the asphalt binder model went through four stages in the simulation of tensile failure, namely “tensile extension”, “nanocrack generation”, “crack adding, expanding and penetrating” and “cracking failure”.(2)Changing the temperature and rate conditions of the asphalt binder model tensile simulation reveals that the higher the tensile temperature, the faster the growth of the crack volume at the beginning of the asphalt binder model tensile, thus reducing its tensile strength; the higher the tensile rate, the weaker the rearrangement of molecules at the gap or crack–separation interface during tensile.(3)By performing a multistage analysis of asphalt binder self-healing after referring to multiple stages of polymer self-healing, the asphalt binder self-healing process is divided into the four mutually overlapping stages of surface approach, surface rearrangement, surface wetting and diffusion. The new stage division is suitable for simulating asphalt binder nanocrack self-healing with low surface fit, which will have a positive impact on future asphalt binder self-healing MD.(4)The self-healing ability of asphalt binder in different degrees of damage was studied from the perspective of energy change. It is shown that van der Waals energy plays a dominant role in the healing process; the influence of the degree of damage on the healing ability of asphalt binder is more obvious in the three stages of surface approach, surface rearrangement and surface wetting, and has less influence on the diffusion stage.

In this study, MD simulated and demonstrated the nanocrack generation process and self-healing behavior of asphalt binder, and these works complement the deficiencies of existing experimental and observational methods at microscopic dimensions such as the nanoscale. This means that MD has the potential to be one of the tools for pavement designs in the future. In this study, uniaxial tensile simulation was used to simulate the failure process of asphalt binder, but the failure of asphalt material is a complex process involving tension, compression, shear, and bending, so future studies should try to use MD to simulate the principles of the dynamic shear rheometer or bending beam rheometer. In addition, due to the time and dimensional scale of MD, few researchers have correlated the actual experimental results with the MD calculation values, so whether there is some kind of conversion relationship between experimental results or experimental conditions that can bridge the gap between experimental and simulated values will be another future research direction.

## Figures and Tables

**Figure 1 polymers-14-03581-f001:**
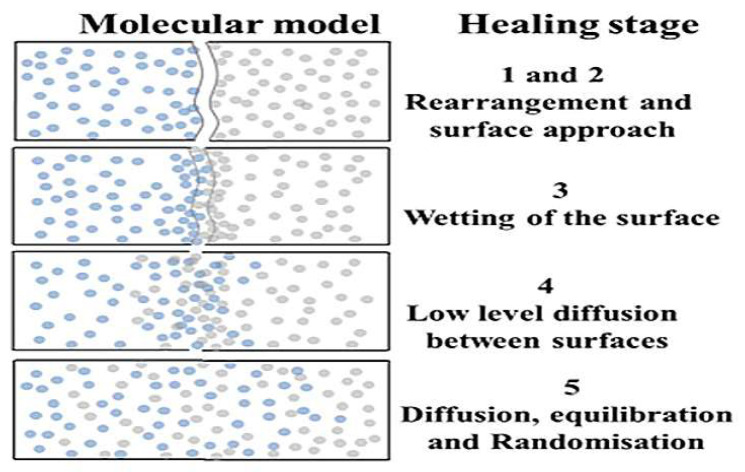
Self-healing process of polymer molecular model. Adapted with permission from [7]. Copyright 2021 Elsevier.

**Figure 2 polymers-14-03581-f002:**
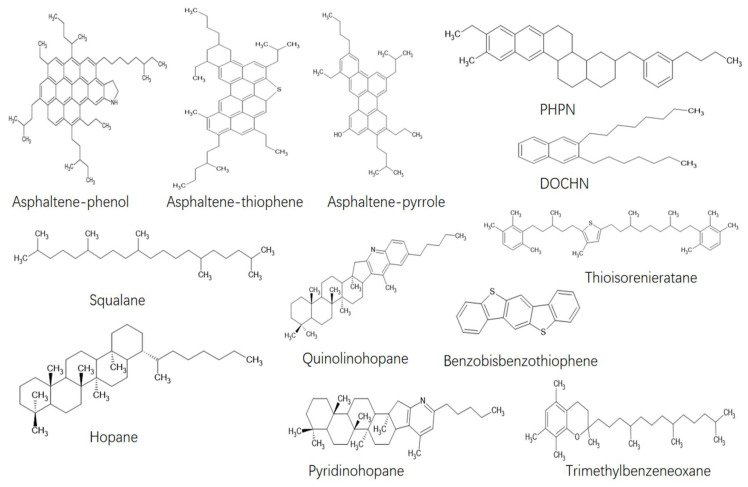
Molecular models of asphalt binder.

**Figure 3 polymers-14-03581-f003:**
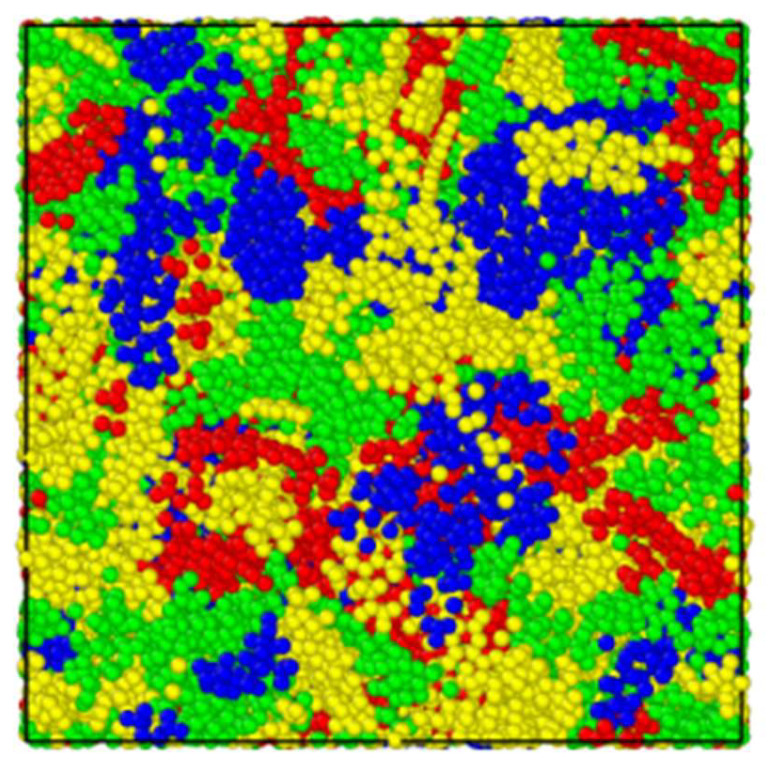
Molecular model of asphalt binder (red: asphaltene, blue: saturated, green: aromatic, yellow: resin).

**Figure 4 polymers-14-03581-f004:**
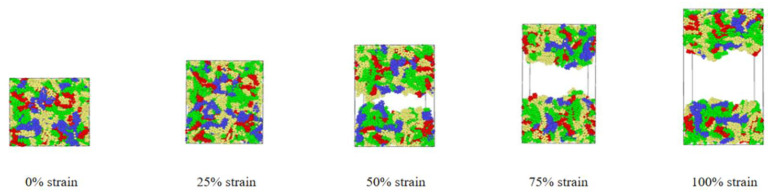
Stretching process under 25 °C, 10 m/s.

**Figure 5 polymers-14-03581-f005:**
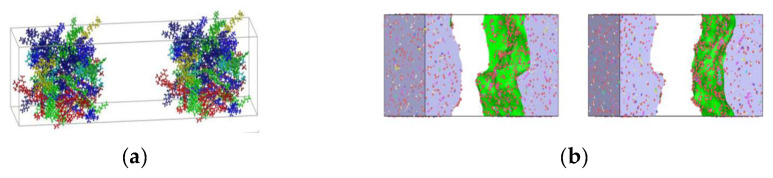
(**a**) The initial model of self-healing of asphalt binder with artificial insertion of vacuum zone. Adapted with permission from [24]. Copyright 2020 Elsevier; (**b**) The initial model of self-healing of asphalt binder selected in this paper.

**Figure 6 polymers-14-03581-f006:**
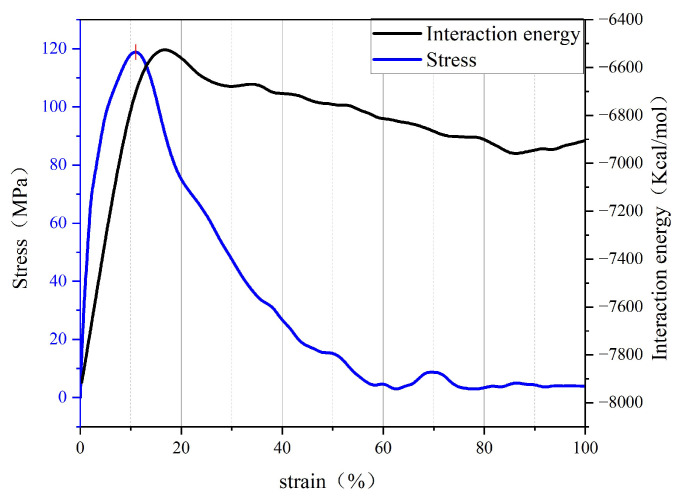
Curves of stress and interaction energy during tensile process.

**Figure 7 polymers-14-03581-f007:**
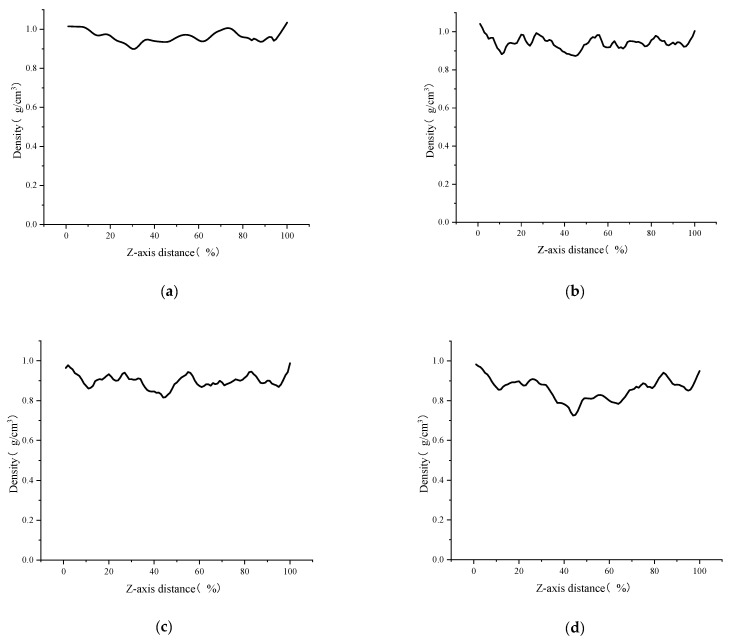
Distribution function of density on the z-axis under different strain conditions. (**a**) 0% strain; (**b**) 5% strain; (**c**) 10% strain; (**d**) 15% strain; (**e**) 20% strain; (**f**) 30% strain; (**g**) 45% strain; (**h**) 60% strain.

**Figure 8 polymers-14-03581-f008:**
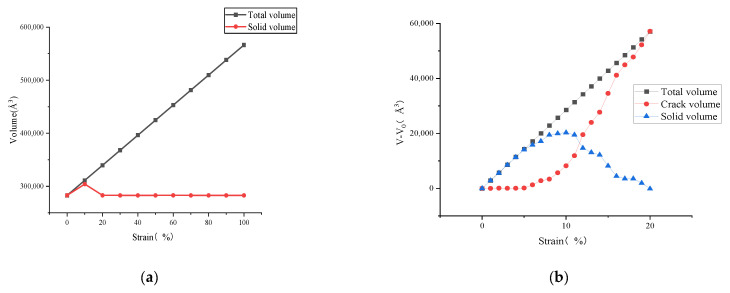
(**a**) Total volume versus asphalt volume change during stretching; (**b**) volume change of each part during 0–20% strain stage.

**Figure 9 polymers-14-03581-f009:**
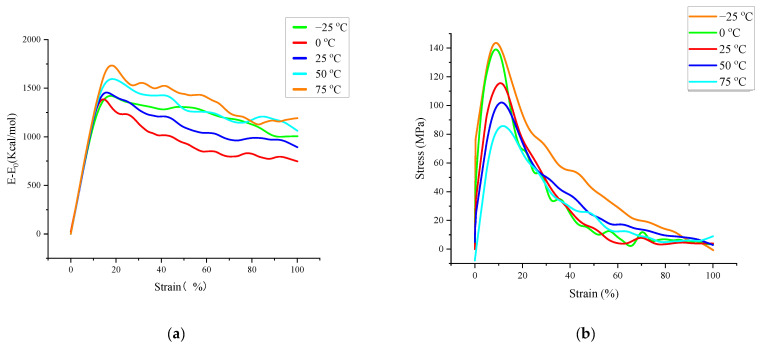
Interaction energy change–strain and stress–strain curves for tensile failure processes under different temperature conditions: (**a**) Interaction energy change–strain curves at different temperatures (**b**) Stress–strain curves at different temperatures.

**Figure 10 polymers-14-03581-f010:**
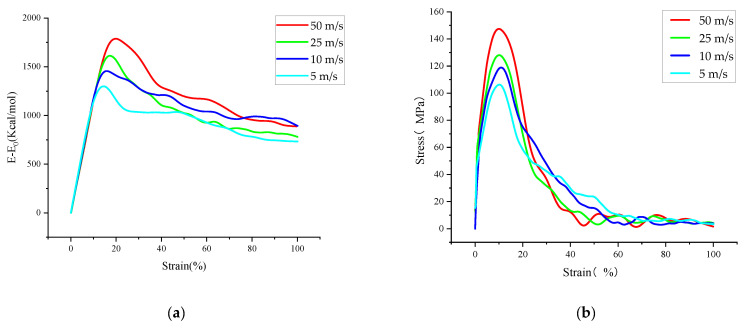
Interaction energy change–strain and stress–strain curves of tensile failure processes under different tensile rates: (**a**) Interaction energy change–strain curves at different rates (**b**) Stress–strain curves at different rates.

**Figure 11 polymers-14-03581-f011:**
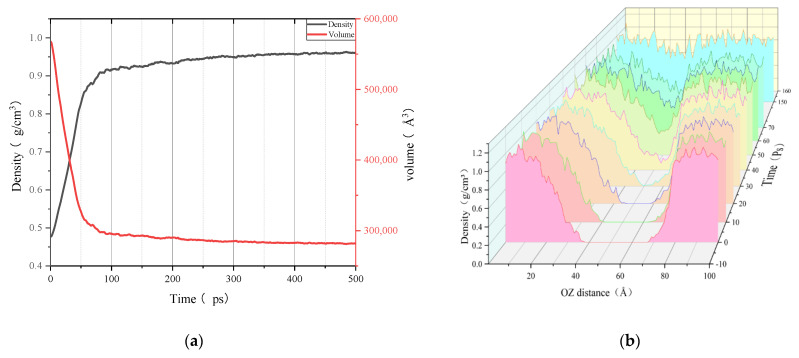
Density and volume change curves during healing process and density distribution curves under different time conditions: (**a**) variation of density and volume with self-healing time; (**b**) density distribution curves under different time conditions.

**Figure 12 polymers-14-03581-f012:**
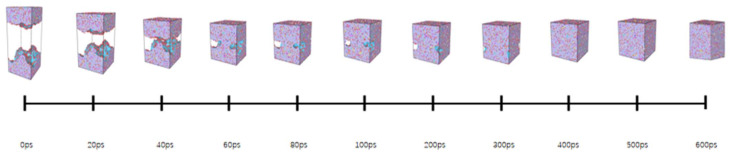
Variation of healing process model.

**Figure 13 polymers-14-03581-f013:**

Crack surface contact picture (blue color is the projection of crack blank area in X-Y section; white area is the asphalt binder fragment surface contact area).

**Figure 14 polymers-14-03581-f014:**
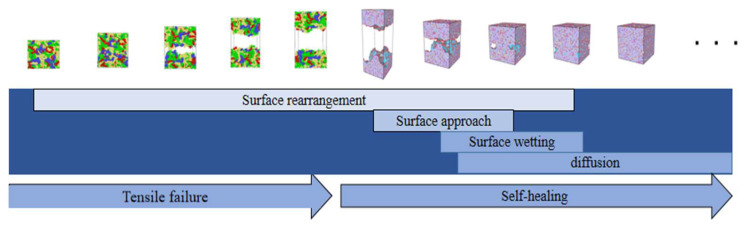
Distribution of self-healing stages in the “stretching-self-healing” process.

**Figure 15 polymers-14-03581-f015:**
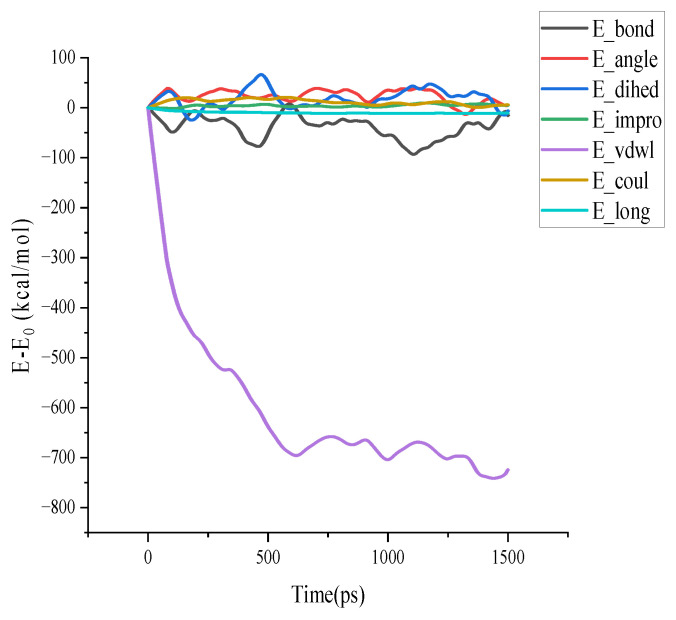
Energy change diagram of self-healing process.

**Figure 16 polymers-14-03581-f016:**
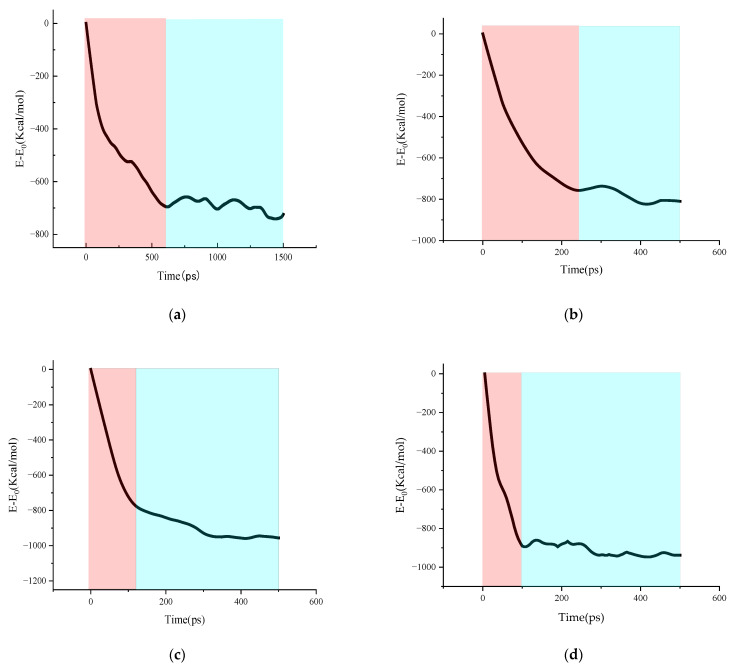
Van der Waals energy change during asphalt binder healing at different damage levels: (**a**) 100% damage; (**b**) 75% damage; (**c**) 50% damage; (**d**) 25% damage (The red area is the curve A area, the cyan area is the curve B area).

**Figure 17 polymers-14-03581-f017:**
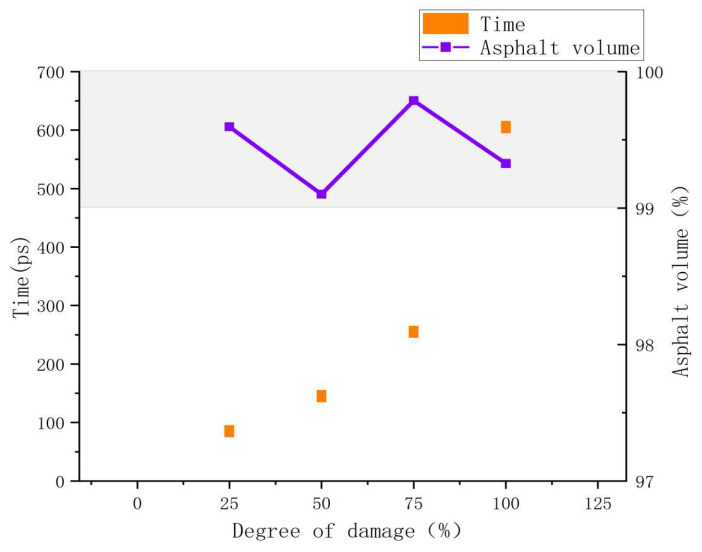
Different damage degree inflection point time and asphalt volume ratio.

**Table 1 polymers-14-03581-t001:** Molecular compositions of asphalt binder model.

Components	Name	Molecular Mass	Number
Asphaltene	Asphaltene-phenol	575.0	3
	Asphaltene-pyrrole	888.5	2
	Asphaltene-thiophene	707.2	3
Saturated	Squalane	422.9	4
	Hopane	483.0	4
Aromatic	PHPN	464.8	11
	DOCHN	406.8	13
Resin	Quinolinohopane	554.0	4
	Thioisorenieratane	573.1	4
	Trimethylbenzeneoxane	414.8	5
	Pyridinohopane	530.9	4
	Benzobisbenzothiophene	290.4	15

**Table 2 polymers-14-03581-t002:** MD simulation steps.

Steps	Time	Timestep	Temperature	Ensemble	Command
1	*	*	*	*	min_style cg
2	50 ps	0.1 fs	600 K	NVT	*
3	100 ps	1 fs	600 K-298.15 K	NVT	*
4	400 ps × 5	1 fs	298.15 K-600 K-298.15 K	NVT	*
5	3 ns	1 fs	298.15 K	NPT	*

* The parameter is not present in the corresponding simulation step.

**Table 3 polymers-14-03581-t003:** Asphalt binder model validation data.

	Calculation	Experiment
Density at 298.15 K (g/cm^3^)	0.981	1.01–1.04 [37]
Glass Transition Temperature (K)	267.7	261.73 [37]
Solubility Parameter ((J/cm^3^)^0.5^)	17.50	13.30–22.50 [37]

**Table 4 polymers-14-03581-t004:** Energy values for different damage levels.

Damage Degree (%)	E (kcal/mol)	E_1_ (kcal/mol)	ΔE_2_ (kcal/mol)	ΔE_3_ (kcal/mol)
25	−7411.2570	−6322.1256	−933.9347	−155.1967
50		−6423.3220	−832.0332	−155.9018
75		−6477.5566	−767.1836	−164.5168
100		−6610.9588	−650.7682	−149.5300

## Data Availability

The data of this study are included in the manuscript.

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
