# Peer review of "A Multistage Analysis of Asphalt Binder Nanocrack Generation and Self-Healing Behavior Based on Molecular Dynamics"

_polymers, 2022, doi:10.3390/polym14173581_

Round 1
Reviewer 1 Report
These are reviewers comments to the manuscript entitled: 'A Multi-Stage Analysis of Asphalt Nano-Crack Generation and Self-Healing Behavior Based on Molecular Dynamics'. The work presented is interesting and definitely timely, when asphalt engineers and scientist are trying to integrate pavement performance models in pavement designs. Unfortunately authors stop short of making this work great by not including experimental (verification) date of their work. Which raises many questions related to the work, such as how accurate are simulation data to the real life material performance. As such I would like to recommend to authors to include experimental verification in the test programme.
Further more authors used to simulate uniaxial tensile test, this is simplest test and fails to perform complex bituminous (asphalt) material behaviour. Can authors explain how will model behave (simulate) fracture if a complex three point bent loading is applied where compressive and tensile stresses are applied. Also, primary test for evaluation of the bitumen would be Dynamic Shear Rheometer or Bending Beam Rheometer, authors should have simulated one of these tests.
More general comments:
Extensive English Language and text formatting review is needed throughout the manuscript, please revise. Eg.: Introduction Ln 1: Asphalt is kind of .... Asphalt is or isn't, it can't be kind of - please rewrite it; 90% of pavement in modern cities is ... - 90% of asphalt pavements in modern cities are ...; ln 37 - 40 is confusing please rewrite it; Fig 1. poor quality and no source reference - please improve the quality and add source reference; pg2 ln 6 - Early researchers .... Researchers at begining or start ....; ln 66 - subsequently large number of studies and tests ... - refrences are missing for these studies please include relevant references; pg 2 ln 74 MD- give full description when first mentioned in text; pg 3 ln 98 remove apostrophe; include full descriptions of acronyms: LAMMPS, VMD, OVITO; pg. 5 ln 180 - 188: simulation steps should be tabulated, the way they are presnted in the text are very difficult to follow.
terminology should be clarified, as I am from Europe when I started reading I have thought that asphalt - means full asphalt mix (aggregates, bitumen, filler), but later I have realised that authors refer to binder or bitumen. Please clarify the terminology.
Abstract, ln 3-5: Formation process of asphalt nanocracks ... I believe that there are numerous theories and studies conducted to date on the crack formation and propagation. Authors give impression that they are coming up with theories for the first time. Please rewrite it.
Abstract, ln 8-9: Asphalt healing is well known theory - even authors discuss it in the introduction, impression given again is that authors are coming up with the theories... please rewrite it.
Authors should use SI units in the manuscript, please use oC instead of K.
When referring to the previous work authors must provide references to the work, eg. pg 7 ln 226
Table 2. experimental data was referenced from 3 different sources (works) considering that bitumen (asphalt) behaviour varies from type to type and source to source how accurate are model results? How authors can be sure that results obtained are correct? As such I am suggesting that authors perform experimental tests and include it into the manuscript.
Reviewer 2 Report
The paper is an interesting and complex study by MD of the self healing process of a model asphalt. The results obtained could be of interest for further studies.
The approach is correct, the abstract is clear but conclusions are missing. The discussion section could be lumped with the results section as a part of the facts mentioned in discussions are already mentioned. Some conclusive discussions can be put in a brief “conclusion” section.
The paper needs a thorough revisions in the way of presentation and editing.
Some observations are:
Line 128 “Asphalt is a complex mixture mainly 128 composed of carbon and hydrogen.....”, should be “ it is a mixture of components containing mainly C and H….”
Line 170 PCFF force field or Pcff forcefield... Always in the same way ( PCFF preferred)
Line 185-189 femtosecond... always fs, not FS, while Kelvin always capital K ( not small letter k) all over the text (influence of temperature)
(as shown in figure.(5)(B)). Observation figure (5) (b) decide:- B or b
Line 273 virgin asphalt and not Virgin asphalt
Line 301 “Figure (7) and Figure (8) show the density distribution curve of the z-axis of the 301 model and the change data of each volume under different strain degrees.”... Totally unclear. How do you define the distribution function of an axis???
It probably is the distribution function of different properties on the z axis (which, according to figures represents a distance or % strain). It must be a distribution function of some properties along z axis of the model... You should write this part correctly, to be understandable
Editing problems : after ; must write with small letter (lines439- 440, ). After . (period) you write with capital letter ( lines393-396). Author’s names must be written with capital letters ( ex: Line 100 Daquan sun... D. Sun, little[13] should be Little[13] , Liang He-line 106 ). Please carefully check all similar cases along the text.
Line 415-417 makes no sense as it has no predicate. Maybe, “thus leading to a later appearance of the interaction energy peak, which ALSO indicates that the energy analysis of the model is more accurate at low loading rates
Lines 431-435, just presenting some figures by writing again what it already is put in figures caption has no sense. May be, before the figure a short comment that figures 11-13 present the evolution of healing process. The comments are presented further in the text and give good insight of the process.
“The diffusion stage, as one of the stages in the self-healing process of asphalt that has the greatest impact on the recovery of material strength”. (this sentence has no predicate) Probably you mean: The diffusion stage, as one of the stages in the self-healing process of asphalt HAS the greatest impact on the recovery of material strength.
The same observation at line 440-443 (“The surface rearrangement stage, which is considered as the process of the slow movement of the crack surface atoms from the unreasonable structure due to tensile damage until the formation of a lower energy structure, and is therefore a stage highly relevant to the existence of cracks in asphalt mode”). Suggestion: The surface rearrangement stage is considered as the process of ....... otherwise you have no predicate in the main sentence.
Round 2
Reviewer 2 Report
The authors made all required corrections and answered to all questions.
The paper may be published in the present form.